

# Sponge exhalent seawater contains a unique chemical profile of dissolved organic matter

Cara L. Fiore[1], Christopher J. Freeman[2] and Elizabeth B. Kujawinski[1]

[1] Marine Chemistry & Geochemistry, Woods Hole Oceanographic Institution, Woods Hole, MA, United States
[2] Smithsonian Marine Station, Smithsonian Institution, Fort Pierce, FL, United States

## ABSTRACT

Sponges are efficient filter feeders, removing significant portions of particulate and dissolved organic matter (POM, DOM) from the water column. While the assimilation and respiration of POM and DOM by sponges and their abundant microbial symbiont communities have received much attention, there is virtually no information on the impact of sponge holobiont metabolism on the composition of DOM at a molecular-level. We applied untargeted and targeted metabolomics techniques to characterize DOM in seawater samples prior to entering the sponge (inhalant reef water), in samples exiting the sponge (exhalent seawater), and in samples collected just outside the reef area (off reef seawater). Samples were collected from two sponge species, *Ircinia campana* and *Spheciospongia vesparium*, on a near-shore hard bottom reef in the Florida Keys. Metabolic profiles generated from untargeted metabolomics analysis indicated that many more compounds were enhanced in the exhalent samples than in the inhalant samples. Targeted metabolomics analysis revealed differences in diversity and concentration of metabolites between exhalent and off reef seawater. For example, most of the nucleosides were enriched in the exhalent seawater, while the aromatic amino acids, caffeine and the nucleoside xanthosine were elevated in the off reef water samples. Although the metabolic profile of the exhalent seawater was unique, the impact of sponge metabolism on the overall reef DOM profile was spatially limited in our study. There were also no significant differences in the metabolic profiles of exhalent water between the two sponge species, potentially indicating that there is a characteristic DOM profile in the exhalent seawater of Caribbean sponges. Additional work is needed to determine whether the impact of sponge DOM is greater in habitats with higher sponge cover and diversity. This work provides the first insight into the molecular-level impact of sponge holobiont metabolism on reef DOM and establishes a foundation for future experimental studies addressing the influence of sponge-derived DOM on chemical and ecological processes in coral reef ecosystems.

Corresponding author
Cara L. Fiore, fiorec@appstate.edu

## INTRODUCTION

Coral reefs exist in relatively oligotrophic environments where there is tight coupling between benthic and pelagic nutrient cycles (*Wild et al., 2008*; *Naumann et al., 2012*). As
both filter feeders and hosts to abundant symbiont communities, sponges can have a major role in coral reef biogeochemistry (*Southwell et al., 2008*) through the removal of particles (*Ribes et al., 2005*) and the transfer of dissolved organic matter (DOM) to higher trophic levels in the benthos (*De Goeij et al., 2013*; *Rix et al., 2016*). Sponge holobiont (sponge plus associated microbes) metabolism has been shown to influence the composition of inorganic nitrogen (e.g., *Southwell et al., 2008*) and the concentration of dissolved organic carbon (DOC; *Ribes, Coma & Gili, 1999*; *Yahel et al., 2003*). However, the impact on the composition of DOM is not well characterized. DOM comprises dissolved organic carbon, nitrogen, phosphorus, and other elements (*Hedges, 2002*), but the carbon component is the largest. Recent studies have begun to examine the sources and major components of DOM on coral reefs. For example, macroalgae generally produce exudates that are higher in neutral sugars and in DOC concentration than coral exudates (*Haas et al., 2011*; *Nelson et al., 2013*). Studies comparing DOC from different sources and their effects on ecosystem function have begun to elucidate ecologically important connections between DOC, microbial community composition, and coral reef biogeochemical cycling (e.g., *Wild et al., 2008*; *Haas et al., 2013*).

The composition of DOM can influence the population structure of microbial communities (*Landa et al., 2013*), including those in coral reef seawater and sediment (*Haas et al., 2013*; *Nelson et al., 2013*). DOM composition also influences coral health (*Morrow et al., 2012*) and ecosystem-wide nutrient cycling (*Wild et al., 2004*). Corals, as well as algae, contribute to DOM by releasing neutral sugars such as fucose and galactose, although the proportion of these components is different between the two sources (*Nelson et al., 2013*). To date, however, our understanding of the composition of coral reef DOM has been limited by methods that cannot fully resolve the complex nature of DOM, particularly within the low-molecular weight component (LMW DOM, <1,000 Da). This fraction could be particularly important to reef ecosystems as it contains molecules that are produced during metabolism and that may be labile to other organisms (e.g., *Mazur & Homme, 1993*; *Amin et al., 2015*).

Sponges are typically considered to be a sink for DOM, releasing exhalent water that has lower concentrations of carbon relative to water entering the sponge (*Wilkinson, 1987*; *Yahel et al., 2003*; *De Goeij et al., 2008b*; *McMurray et al., 2016*). It is thus not surprising that DOM is an important carbon source for many sponges, contributing up to 10% of total carbon assimilated by two deep water sponges (*Van Duyl et al., 2008*) and contributing over an order of magnitude more carbon to the biomass of coral reef sponges than POM (*Yahel et al., 2003*). The cryptic reef framework, where sponges are common, has also been shown to be a major sink for DOM (*Van Duyl & Gast, 2001*; *De Goeij & Van Duyl, 2007*), most of which is likely assimilated by sponges (*De Goeij et al., 2008b*). Additionally, coral-excavating sponges rely largely on DOM to meet their carbon demand and may be able to take advantage of nutrient-rich environments (*Mueller et al., 2014*), highlighting the pivotal role of DOM in these ecosystems.

While the removal of DOM by sponges has been documented for some time (*Stephens & Schinske, 1961*; *Reiswig, 1990*; *Alber & Valiela, 1995*; *Ribes, Coma & Gili, 1999*), it was unclear until recently whether sponge cells could assimilate DOM directly or only *via*

their microbial symbionts (*sensu De Bary, 1879*). An experiment with isotopically labeled DOM and particulate organic matter (POM) demonstrated that both sponge-associated microbes and the sponge itself are capable of assimilating dissolved material (*De Goeij et al., 2008a*; *De Goeij et al., 2013*), although the specific components of DOM removed by the sponge holobiont could not be determined. A few studies have provided preliminary insight into these components, where the uptake of several amino acids, but only a minimal amount of glucose, by sponges was observed (*Stephens & Schinske, 1961*; *De Goeij et al., 2008a*). The difference in uptake between the amino acids and glucose may indicate some discrimination in the removal of metabolites from seawater by the sponge holobiont, consistent with previous observations of differential metabolite removal by marine free-living microbes (*Malmstrom et al., 2004*; *Nelson & Craig, 2012*).

Sponges, with their high abundance on many reefs, particularly in the Caribbean (*McMurray, Finelli & Pawlik, 2015*), their large filtration capacity (upwards of 30 $L^{-1}$ $hr^{-1}$ per L of sponge; (*Weisz, Lindquist & Martens, 2008*)) and microbial symbiont diversity (e.g., *Easson & Thacker, 2014*), likely influence the composition and availability of DOM on coral reefs. Here we use high resolution untargeted and targeted metabolomics techniques to probe the composition of DOM on a near shore reef in the Florida Keys (USA) and to characterize the influence of the sponge holobiont metabolism on DOM composition. Untargeted, or discovery-based, analysis allows for a semi-quantitative profile-view of low molecular weight molecules within a sample, while the targeted metabolomics approach provides a quantitative comparison of molecules relevant to the growth of many organisms (*Patti, Yanes & Siuzdak, 2012*; *Kido Soule et al., 2015*). In the present study, two sponge species were selected for analysis that are known to harbor distinct symbiont communities: *Ircinia campana*, a high microbial abundance (HMA) sponge and *Spheciospongia vesparium*, a low microbial abundance (LMA) sponge (*Poppell et al., 2014*). By applying metabolomics analysis to inhalant and exhalant water of these two sponge species and to water collected away from the reef, we demonstrate a complex dynamic of removal and addition of labile DOM to the surrounding seawater that would be difficult to predict based solely on bulk carbon analyses

## MATERIALS AND METHODS

### Sample collection

Seawater samples were collected near Long Key State Park, FL (24.8169°N 80.8200°W) by scuba divers (4–5 m depth; Fig. 1). Abundance data on sponge populations were not obtained, but in general sponges were in relatively low abundance compared to populations typical of offshore coral reefs in the Florida Keys. Acid-washed polypropylene syringes (100 ml) attached to 1 L FlexFoil Plus sample bags (SKC, Pittsburgh, PA, USA) *via* a three-way valve with Teflon tubing were used to collect ambient reef water and exhalant water from the sponge. The valve allowed the syringe to be filled and then pumped into the bag ten times to collect 1L of seawater. Ambient reef water (herein referred to as inhalant) was collected near each sponge but away from the direction of the exhalant plume. Exhalant water from each sponge was collected by drawing water into the syringe slowly (1 ml $s^{-1}$)
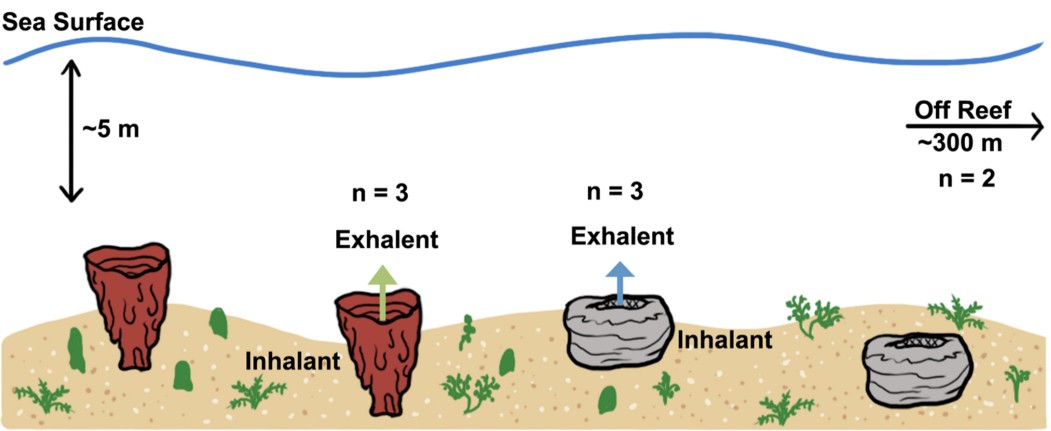

**Figure 1 Schematic of sampling location and sample types.** The three sample types are shown: inhalant, exhalent, and off reef seawater. The red vase-like sponges represent *Ircinia campana* and the gray sponges represent *Spheciospongia vesparium*. The numbers of samples collected for each sample type are shown. For inhalant and exhalent samples, one sample type was collected from each of three individuals. Reef seawater samples were collected near the surface approximately 300 m from the reef area. Arrows represent direction of water flow and color is associated with samples representing each sponge species in later figures.

through tubing placed just above the osculum for *Spheciospongia vesparium* and just above the base of the spongeocoel for *Ircinia campana*. All sponges (three individuals per species) were checked for pumping activity using fluorescein dye and sampled once the dye had visibly disappeared. The time the dye front took to move from the base of the spongeocoel to the osculum was recorded in order to obtain the centerline fluid velocity and to estimate pumping rates. While this method can be confounded by sponge morphology (*McMurray, Pawlik & Finelli, 2014*), it has been shown to provide a rough estimate of pumping rate at least within species (*Weisz, Lindquist & Martens, 2008*; *Southwell et al., 2008*; *Fiore, Baker & Lesser, 2013*). Thus, we used pumping rates to provide an order of magnitude estimate for the release of metabolites to the reef. Volume measurements described previously (*McMurray, Blum & Pawlik, 2008*) were recorded for each sponge using measuring tape (to ±1.0 mm) to calculate sponge volumes (*McMurray, Blum & Pawlik, 2008*; *Weisz, Lindquist & Martens, 2008*). The height of the sponge was assumed to be the height of the spongeocoel for *S. vesparium*. Off reef samples were collected in polycarbonate bottles ($n = 2$, 1.5 L total) by hand near the surface and approximately 300 m seaward from the reef, over a mix of sand and hard substrate.

The seawater was put on ice immediately upon end of the dive. Upon return to the Smithsonian Marine Station (Fort Pierce, FL), subsamples (40 ml) of each water sample were added to combusted vials and acidified to pH = 2 using 12 N hydrochloric acid, and stored frozen for total organic carbon (TOC) analysis. TOC concentrations were used to calculate the extraction efficiency for a subset of samples. The rest of the seawater was filtered through a combusted glass fiber filter (GF/F; 0.7 µm; Whatman) and a 0.2 µm PTFE filter (Omnipore, Millipore, MA, USA). The filtrate was frozen and shipped to Woods Hole Oceanographic Institution (WHOI, Woods Hole, MA, USA) for extraction of dissolved organic matter.

## Extraction of low molecular weight dissolved organic matter and instrument methods

Extractions of filtrate samples and the instrument methods used here were performed as described in *Kido Soule et al. (2015)* and *Fiore et al. (2015)*. Briefly, the seawater filtrate was acidified to pH 3 (*Longnecker, 2015*) and then de-salted and extracted with1g/6 cc PPL solid phase extraction cartridges (BondElut; Agilent, Santa Clara, CA, USA) as described previously. DOM was eluted from the column using 100% methanol prior to storage at −20 °C until analysis. Just prior to mass spectrometry analysis on the instruments described below, the methanol extract was dried down and redissolved in 500 μl of 95:5 water:acetonitrile. Deuterated biotin was added to each sample as a high-performance liquid chromatography (HPLC) injection standard (final concentration 0.05 μg ml$^{-1}$). At this stage, 100 and 200 μl of the extract was removed for targeted and untargeted metabolomics analysis respectively. Equal amounts of each experimental extract were combined to create a pooled sample.

Samples for TOC analysis, once reaching WHOI, were stored at 4 °C until analysis with a Shimadzu TOC-V$_{CSH}$ total organic carbon analyzer (*Hansell & Carlson, 2001*). MilliQ water blanks, standard curves using potassium hydrogen phthalate, and certified standard reference material (the latter provided by Prof. D Hansell, University of Miami) were used for instrument calibration and were made fresh each day. Targeted metabolomics analysis was performed using high-performance liquid chromatography (Thermo PAL autosampler and Accela pump) coupled to a triple stage quadrupole mass spectrometer (TSQ Vantage, Thermo Fisher Scientific, MA, USA) via a heated electrospray ionization (H-ESI) source operated in both positive and negative ion modes. A set of standards, consisting of amino acids, vitamins, nucleosides, and other metabolites (*Kido Soule et al., 2015*), was used in the targeted method to identify and quantify these compounds in the experimental samples. Some metabolites have low recovery yields following SPE (WM Johnson, MC Kido Soule, EB Kujawinski, 2016, unpublished data), thus a subset of 23 standards that have high recovery rates with SPE were analyzed in the experimental samples. Selected reaction monitoring (SRM) conditions were optimized for each standard.

Untargeted analysis of extracted metabolites was performed using high-performance liquid chromatography (Micro AS autosampler and Surveyor MS pump plus) coupled *via* ESI to a hybrid linear ion trap-Fourier transform-ion cyclotron resonance (FT-ICR) mass spectrometer (7T LTQ FT Ultra; Thermo Fisher Scientific). Electrospray ionization is a soft ionization technique and, as such, minimizes molecular fragmentation in the source. Thus, each mass feature likely represents an individual compound. The pooled sample was analyzed every seven injections in both the targeted and untargeted metabolomics methods for quality control. More details on instrument methods have been previously described (*Kido Soule et al., 2015*; *Fiore et al., 2015*), with the exception that only positive ion mode for the untargeted metabolomics method was analyzed in the current study

## Metabolomics data analysis

Untargeted metabolomics data were manually inspected in XCalibur (2.0) for issues with injection, contamination, or ion suppression by comparing peak intensities of the injection
standard in each sample, and peak intensities of the pooled samples prior to performing analysis in XCMS. Peak intensities did not change substantially over the time course of the analytical run so it is unlikely that there was any systematic decrease in instrument response over time. Manual inspection of chromatograms in XCalibur did reveal contamination by plastics (e.g., polyethylene glycol) in all samples, easily distinguished by regularly spaced mass features (44 $m/z$). These mass features were present from 9.3 to 11.3 min; therefore, all features within this time frame were removed (263 mass features; 8%). The untargeted metabolomics data were processed with XCMS 1.38.0 (*Smith et al., 2006*) and CAMERA 1.18.0 programs (*Kuhl et al., 2012*) in R. The resulting peak table from XCMS and CAMERA was further qualitychecked by retaining only mass features that were present in at least two out of the three pooled samples and removing features within 10 ppm of fluorescein (as $[M + H]^+$; 4 features). Typical mass errors are 1–2 ppm, making this a very conservative criterion for fluorescein. Mass features are defined here as a unique combination of mass-to-charge ($m/z$) value and retention time. Peak areas of mass features are proportional to metabolite concentration (e.g., *Fiore et al., 2015*) and thus were normalized to the filtrate volume for analysis. Mass features of interest that had an associated fragmentation spectrum were processed using the xcmsFragment function in XCMS (*Benton et al., 2008*) and the spectra were queried against the METLIN metabolomics database (*Smith et al., 2005*) and in the program MetFrag (*Wolf et al., 2010*) as described in *Fiore et al. (2015)*. All metabolomics data are available through the MetaboLights online database (http://www.ebi.ac.uk/metabolights/) by accession number MTBLS281.

Statistical analyses were performed with R statistical software (v3.0.2; R Core Team). The function metaMDS (vegan package; *Oksanen, 2016*) was used to perform non-metric multidimensional scaling (nMDS) of volume-normalized untargeted data. The dimensionality was assessed by Monte Carlo analysis using 20 iterations with real data and 20 iterations with randomized data for each of several dimensions ($n = 2$–5). Mantel tests were performed to determine the coefficient of determination ($r^2$) between ordinal distance of the nMDS and the Bray-Curtis dissimilarity for both axes and each axis separately. Analysis of similarity (ANOSIM) was performed to test for significant differences in the diversity and abundance of mass features between groups detected in the nMDS. A paired $T$-test was used to compare peak areas between all inhalant ($n = 6$) and all exhalent ($n = 6$) seawater samples with log transformation to meet test assumptions (peaks required to be in five out of six replicates, any of which could have a missing sample). A Welch's $T$-test was used to compare exhalent seawater samples by species (log transformed; peaks required to be in all three replicates per species) The generated $p$-values were adjusted for multiple comparisons using false discovery rate (FDR; *Benjamin & Hochberg, 1995*).

Targeted metabolomics analysis was performed using five-to-seven point manually-curated external calibration curves for each standard (0.5–500 ng ml$^{-1}$) to determine relative metabolite concentrations (XCalibur 2.0). The concentrations were then exported to Microsoft Excel. A paired $T$-test, followed by FDR $p$-value corrections, was also used to test for significant differences in log-transformed metabolite concentrations between the exhalent and inhalant reef seawater samples within each species ($n = 3$ per species). Additionally, metabolite concentration data for the two sponge species were pooled to test

for differences in concentration between all exhalent and all inhalant seawater samples ($n = 6$ inhalant and $n = 6$ exhalent), using a paired $T$-test (log transformed) and $p$-values were adjusted as described above. NMDS and ANOSIM were also used, as described for the untargeted metabolomics dataset, to test the effect of sample type (inhalant, exhalent, and off reef) on metabolite composition from the targeted metabolomics dataset. Vectors, representing correlations ($n = 1,000$ permutations) between metabolite concentrations and sample type (inhalant, exhalent, off reef), were added to the plot using the function envfit, part of the Vegan package in R. The length of the vector corresponds to the strength of the correlation with a longer vector indicating a stronger correlation.

The amount (pmol) of a given metabolite added to the reef seawater from a sponge over a 24-hour day was calculated by multiplying the difference in concentration of the metabolite in the exhalent and inhalant seawater (pmol L$^{-1}$) by the minimum and maximum range of the volumetric flow rate of the sponges (average = 0.3 L s$^{-1}$ ± 0.28) then multiplied by the conversion of seconds to 24 h. We assume that the sponge continues to pump over a 24-hour period. While sponges do cease pumping, this is highly sporadic occurring anywhere from 2 to 21 days and lasting from 10 min to multiple days depending on the species (*Reiswig, 1971*; *Vogel, 1974*; *Pile et al., 1997*; *McMurray, Pawlik & Finelli, 2014*). Thus, in one 24-hour period it is likely that these sponges would continue to filter and pump water.

## RESULTS

### Metabolic profiles of ambient reef water, sponge exhalent water, and off reef water

Volume normalized pumping rates were 0.17 L s$^{-1}$ L$^{-1}$ sponge (±0.11) for *S. vesparium*, with an average volume of 2.4 L (±0.7) and 0.08 L s$^{-1}$ L$^{-1}$ sponge (±0.12) for *I. campana*, with an average volume of 15.6 L (±15). Extraction efficiency for the SPE-extracted DOM was calculated for several representative samples (IC-A1, SV-A2, SV-E1, SV-E2, Off reef 1, Off reef 2) and ranged from 23 to 59%.

Untargeted metabolomics analysis yielded a total of 2,973 mass features across samples. There was a large overlap in features across all samples, with about half of the features shared between the three sample types (58%). The sponge exhalent water had the highest number of unique mass features (334 features, 12%), followed by off-reef water (195 features, 7%) and inhalant reef water (73 features, 3%). The exhalent seawater and off reef seawater shared the most features (347, 12%), while the exhalent and inhalant seawater samples shared the fewest features (42, 2%). Analysis of similarity following nMDS visualization indicated a significant difference between the sample groups of inhalant and off reef water combined relative to exhalent water (Fig. 2; ANOSIM, $R = 0.398$, $p = 0.016$).

Of the 2,003 mass features present in at least five out of six replicates of sponge exhalent and inhalant water, 243 (12%) were significantly more abundant in the exhalent water ($T$-test, FDR adjusted $p < 0.05$). In contrast, 12 features (<1%) were significantly more abundant in inhalant seawater than in exhalent water ($T$-test, FDR adjusted $p < 0.05$).

Because the off reef samples ($n = 2$) could not be statistically compared to the other sample types, a threshold of 20× the average peak area was delineated to determine

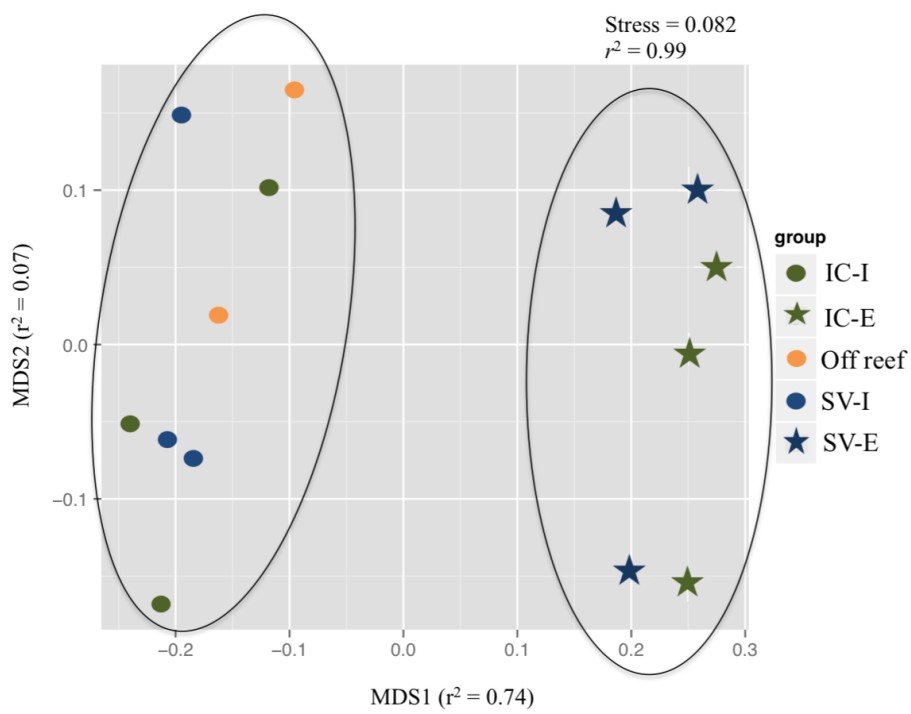

**Figure 2  Non-metric multidimensional scaling (nMDS) plot of dissolved organic matter profiles from untargeted metabolomics analysis.** The strength of the correlation between the distance matrix and the nMDS plot distance are given for each axis and the overall plot. Samples are *Ircinia campana* (IC), *Spheciospongia vesparium* (SV), and off reef. Two groups were distinguished based on analysis of similarity (ANOSIM, $R = 0.398$, $p = 0.016$), which are highlighted by the black ovals. Off reef and inhalant (I) seawater samples grouped together on the left and the exhalent (E) seawater samples grouped together on the right.

whether a peak was higher in a particular sample type and then used for comparative analyses (Table 1). A similar number of differential mass features were yielded using a threshold of 20× difference in peak area and statistical analysis ($T$-test, $p < 0.05$, 430 features greater in the exhalent than inhalant samples, 52 greater in the inhalant samples), to compare the inhalant and exhalent seawater samples. Using this threshold, the off reef samples were first compared to all of the inhalant water samples ($n = 6$). Many features ($n = 106$) were elevated in the inhalant seawater relative to the off reef samples. Interestingly, within these features that are abundant in the ambient reef water, nearly half ($n = 42$) were significantly higher in the exhalent than in the inhalant water ($T$-test, FDR adjusted $p < 0.05$). In contrast, only three features were significantly higher in the inhalant seawater ($T$-test, FDR adjusted $p < 0.05$). There were also 71 mass features that were lower in the inhalant seawater compared to the off reef samples. Of these features, 32 were significantly different between the exhalent and inhalant seawater, all of which were higher in the exhalent seawater samples. The off reef samples were then compared to the exhalent samples, yielding many features that were more abundant in the exhalent seawater samples, particularly from *S. vesparium* (Table 1). Mass features of interest (i.e., those that were more abundant in a certain sample type as described above) with associated fragmentation

**Table 1  Mass features that differentiate different sample types (off reef, inhalant, exhalent) for each sponge species (*Ircinia campana* (IC) and *Spheciospongia vesparium* (SV)).** To determine whether a peak was higher in a particular sample type, a threshold of 20× (average peak area) was used for comparison because the off reef seawater samples have two replicates and thus could not be compared with conventional statistics. Numbers represent features that are greater in the row sample than in the column sample (for instance there were 164 mass features that were, on average, at least 20× higher in the IC inhalant samples than in the off reef samples). 'n.d.', not determined.

|            | Off reef | IC-Inhalant | SV-Inhalant | IC-Exhalent | SV-Exhalent |
|------------|----------|-------------|-------------|-------------|-------------|
| Off reef   | –        | 156         | 164         | 81          | 55          |
| IC-Inhalant | 164     | –           | n.d.        | 27          | n.d.        |
| SV-Inhalant | 155     | n.d.        | –           | n.d.        | 56          |
| IC-Exhalent | 601     | 584         | n.d.        | –           | 93          |
| SV-Exhalent | 587     | n.d.        | 608         | 26          | –           |

spectra were used to query metabolomics databases. None of the mass features investigated here revealed a significant match to any characterized metabolites.

The overwhelming majority of mass features ($n = 2{,}243$, 85%) were detected in both exhalent sample groups (*I. campana* exhalent seawater, IC-E and *S. vesparium* exhalent seawater, SV-E). Of these shared features, $T$-tests revealed no significant differences ($p < 0.05$) between the two species following $p$-value adjustment. When all features present in the exhalent seawater samples in each species were compared ($n = 2{,}623$ features), 121 features had $p$-values below 0.05; however, none of these were significantly different between inhalant and exhalent waters following $p$-value corrections. A number of features were unique to IC-E and SV-E (156 and 227, respectively); however, some of these features were highly variable in their corresponding samples, yielding high $p$-values in statistical comparisons.

## Targeted metabolomics analysis

Sixteen (of 23 possible) metabolites were detected and quantified in these samples, including several amino acids, B vitamins, and nucleosides (Figs. 3 and 4). Most of the 16 metabolites were detected in all sample types; however, the nucleoside xanthosine was only detected in the off-reef samples. Seven of the 16 metabolites were highest in the off reef samples, including some amino acids and caffeine, but also the glycerol derivative, glycerol-3-phosphate (Fig. 3). Within the reef samples (inhalant and exhalent), the exhalent water contained higher concentrations of metabolites than inhalant seawater for several metabolites including most of the nucleosides and the amino acid tryptophan (Fig. 4). Statistical comparison of metabolite concentrations in inhalant and exhalent samples yielded no significant differences after $p$-value corrections ($T$ test, $p > 0.05$ for all). Sample types (inhalant, exhalent, off reef) were significantly different from each other as detected by ANOSIM ($R = 0.42$, $p = 0.004$). The nMDS visualization and vector overlays showed significant correlations between off reef seawater samples and aromatic amino acids, caffeine, and xanthosine, while most of the nucleosides and riboflavin were significantly correlated with the exhalent seawater samples (Fig. 5; Table S1; all $p$-values < 0.05). The inhalant samples, however, were not significantly correlated to any of the metabolites

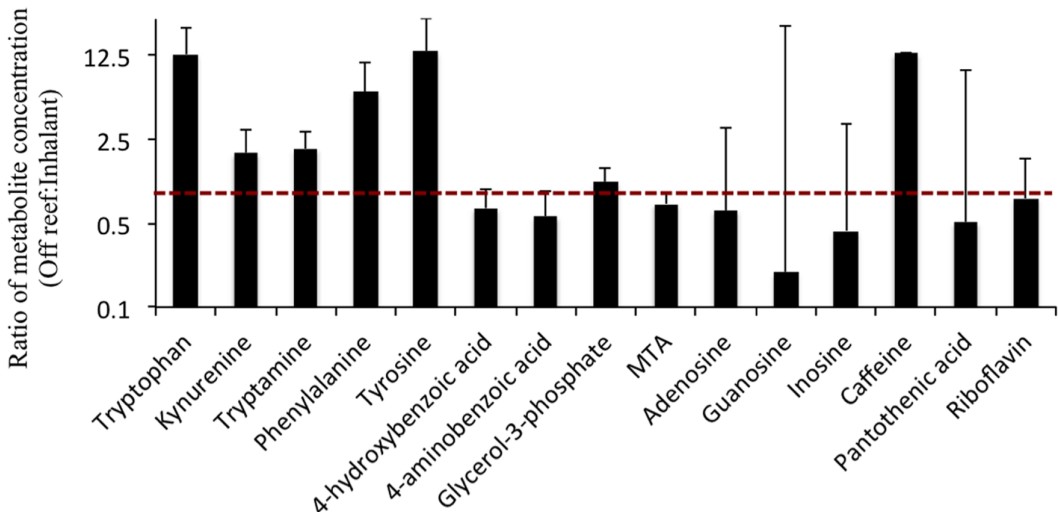

**Figure 3 Average (±SD) ratios of metabolite concentrations of off reef and inhalant seawater samples (off reef/inhalant).** Note that the $y$-axis is presented as log2 scale to highlight equivalent concentrations (ratio = 1; red dashed line). The ratio of each metabolite was calculated by averaging individual ratios of each of the two off reef and each inhalant sample (12 ratios). MTA, 5′-methylthioadenosine. Xanthosine is not shown as it was only detected in the off reef samples.

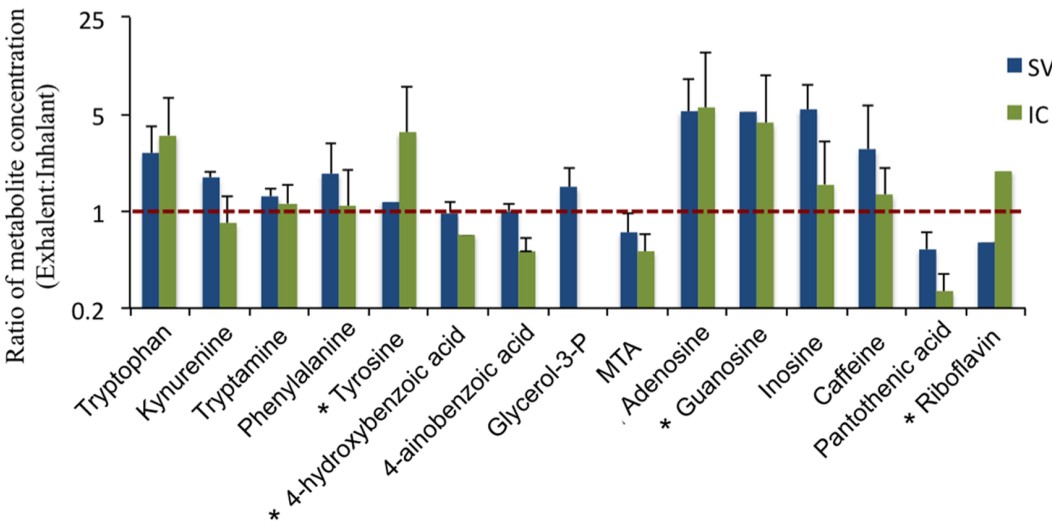

**Figure 4 Average (±SD) ratio of metabolite concentration of exhalent and inhalant seawater samples (exhalent/inhalant) for each sponge species (IC, *Ircinia campana*; SV, *Spheciospongia vesparium*).** Note that the $y$-axis is presented as a log2 scale to highlight equivalent concentrations (ratio = 1; red dashed line). No significant differences were observed between the concentrations of metabolites in the inhalant and exhalent seawater samples. Asterisks indicate metabolites that were detected in only one or two replicates of the inhalant samples resulting in no standard deviation for that metabolite ratio. MTA, 5′-methylthioadenosine. $n = 3$.

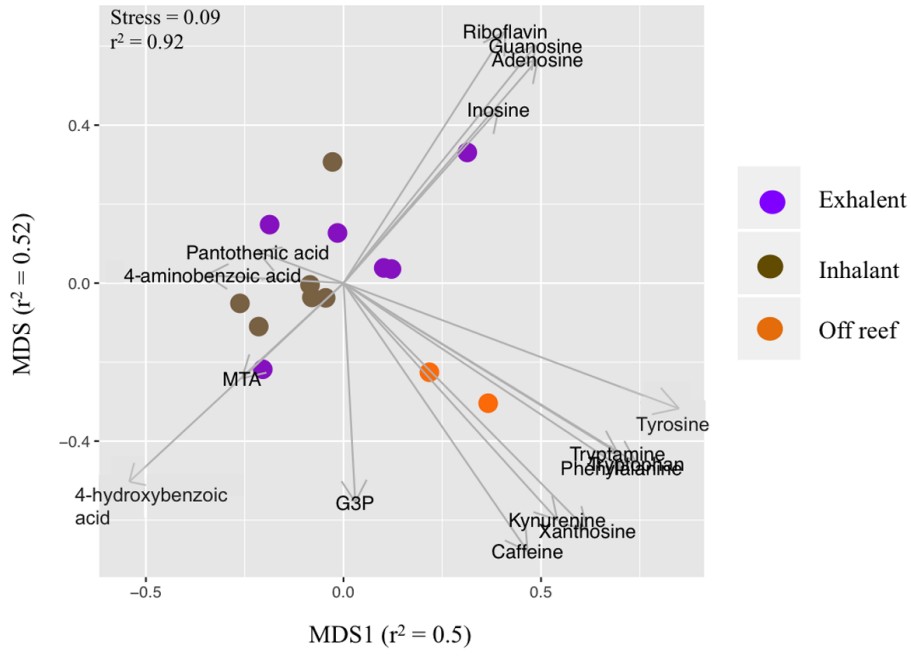

**Figure 5  Non-metric multidimensional scaling (nMDS) plot of metabolite concentrations based on targeted metabolomics analysis** The strength of the correlation between the distance matrix and the nMDS plot distance are given for each axis and the overall plot. Samples are exhalent (for both sponge species), inhalant (for both sponge species), and off reef. These three groups were distinguished based on analysis of similarity (ANOSIM, $R = 0.42$, $p = 0.004$). Vector overlays are based on correlations between metabolite concentration and sample type (inhalant, exhalent, off reef) with a longer vector indicating a stronger correlation. Correlations with pantothenic acid, MTA, and 4-aminobenzoic acid were not significant. G3P, glycerol-3-phosphate, MTA, 5′-methylthioadenosine.

including pantothenic acid in the broader nMDS analysis (Fig. 5; Table S1; $p < 0.05$). Over 24 h, with constant sponge pumping, ∼2 pmol to ∼1 nmol of the metabolites quantified in the targeted method would be released to the water column from each sponge. However, four metabolites (4-hydroxybenzoic acid, glycerol-3-phosphate, 5′-methylthioadenosine, and pantothenic acid) were likely removed by the sponge holobiont, as they were depleted in concentration in the exhalent seawater relative to the surrounding seawater (Table 2). One metabolite, the folate precursor, 4-aminobenzoic acid yielded no net addition or removal to the surrounding seawater as the average concentration in the exhalent and inhalant samples were approximately equal (Table 2).

## DISCUSSION

The present study supports three main conclusions regarding the alteration of DOM by the sponge holobiont: (1) the profile of DOM in the exhalent water is unique from the surrounding seawater and seawater further away from the reef; (2) the influence of sponge-derived DOM on the DOM profile of surrounding seawater is localized close to the exhalent plume; and (3) the DOM profiles of two sponge species with distinct microbial communities are highly similar. The latter conclusion may indicate a characteristic "sponge holobiont signal" for sponge-derived DOM, and provides a testable hypothesis for future work.

**Table 2 Estimated release of metabolites from sponges based on targeted metabolomics analysis.** The average concentration of each metabolite in the inhalant and exhalent seawater with standard deviation is shown. The average concentration in the exhalent water and the minimum and maximum pumping rate (average = 0.3 $L^{-1} \pm 0.28$) was used to estimate the amount of metabolite released per sponge over 24 hours. MTA, 5'-methylthioadenosine.

| Metabolite | Inhalant (fmol $L^{-1} \pm$ SD) | Exhalent (fmol $L^{-1} \pm$ SD) | Min (pmol $day^{-1}$) | Max (pmol $day^{-1}$) |
|---|---|---|---|---|
| Tryptophan | 18 ($\pm$9) | 50 ($\pm$30) | 55 | 1,604 |
| Kynurenine | 7 ($\pm$2) | 8 ($\pm$3) | 2 | 50 |
| Tryptamine | 15 ($\pm$3) | 18 ($\pm$5) | 5 | 130 |
| Phenylalanine | 70 ($\pm$21) | 95 ($\pm$57) | 43 | 1,253 |
| Tyrosine | 4 ($\pm$4) | 12 ($\pm$12) | 14 | 400 |
| 4-hydroxybenzoic acid | 65 ($\pm$36) | 53 ($\pm$26) | −21 | −601 |
| 4-aminobenzoic acid | 2 ($\pm$0.5) | 2 ($\pm$0.6) | 0 | 0 |
| Glycerol-3-phosphate | 13 ($\pm$6) | 10 ($\pm$12) | −5 | −150 |
| MTA | 7 ($\pm$2) | 4 ($\pm$0.8) | −5 | −150 |
| Adenosine | 6 ($\pm$2) | 10 ($\pm$12) | 7 | 200 |
| Guanosine | 3 ($\pm$2) | 19 ($\pm$25) | 28 | 802 |
| Inosine | 6 ($\pm$2) | 20 ($\pm$16) | 24 | 702 |
| Caffeine | 8 ($\pm$3) | 17 ($\pm$20) | 16 | 451 |
| Pantothenic acid | 23 ($\pm$5) | 9 ($\pm$6) | −24 | −702 |
| Riboflavin | 2 ($\pm$1) | 3 ($\pm$2) | 2 | 50 |

Several lines of evidence support the distinction of sponge-derived DOM profiles from those of surrounding seawater. The analysis of similarity based on the untargeted metabolic profile of the seawater samples indicated that the exhalent water samples were significantly different from both the inhalant seawater and off reef seawater DOM profiles. Because the majority of mass features were common to all sample types, this suggests that differences in the concentration of individual compounds are driving the observed statistical separation in samples. However, it is likely to be a combination of diversity and concentration differences in each sample type that are responsible for the statistical observations. For example, there are 340 unique features in the exhalent samples, a small portion of the overall number of features, but these unique features may have a role in driving differences between sample types. Similarly, pairwise comparisons between the shared features of exhalent and inhalant seawater indicated that over 400 metabolites exhibited significantly different concentrations between these two water types. The majority of these differential features were elevated in concentration in the exhalent water, indicating significant contribution of metabolites to the seawater within the immediate vicinity of the sponge. Similarly, many mass features were enriched in the exhalent samples relative to the off reef seawater samples. However, the lack of separation of inhalant and off reef seawater samples based on DOM composition implies that the impact of the distinct exhalent profile is spatially limited as it may be rapidly utilized or diluted.

While there is not a strong impact of sponge-derived DOM on the overall reef DOM profile, the distinct profile of DOM in the exhalent plume of the sponges may influence the metabolism, and potentially community structure of the nearby pelagic microbial community (*Judd, Crump & Kling, 2006*; *Landa et al., 2013*). Given the connections
between the microbial community composition and activity, biogeochemical cycling, and coral health (e.g., *Dinsdale et al., 2008*; *Kline et al., 2006*; *Haas et al., 2011*), it is possible that sponge-derived DOM could also impact the coral reef community as a whole. This work provides another avenue for investigating the dynamics of coral reef microbial communities and the interactive effects of microbial and benthic invertebrate metabolism on overlying seawater chemistry.

Part of the impact of sponge-derived DOM on seawater chemistry was estimated based on the quantified metabolites. Our calculations indicated that over a 24 hr period, these sponges would release between ∼2 picomoles to about a nanomole or more of many of the metabolites quantified using targeted metabolomics analysis. In some cases, the sponges were a net sink for compounds, including 4-hydroxybenzoic acid and pantothenic acid, supporting the idea that sponges are removing certain compounds and adding other compounds to the seawater as it passes through the sponge. As a comparison, total dissolved free amino acids (DFAA) have been quantified previously in Biscayne Bay FL (USA), where their concentrations ranged from ∼15 to 50 nM in the surface waters between 1.5 and 10 m from shore (*Lee & Bada, 1977*). DFAA are typically a small portion of the total dissolved amino acid pool (*Lee & Bada, 1977*; Sommerville & Preston, 2001), but are likely to be labile to the nearby microbial community (*Jørgensen, 1987*; *Middelboe, Borch & Kirchman, 1995*). We estimate that up to a nanomole each of tryptophan and phenylalanine would be released from each sponge each day, potentially making these sponges an important source of these two amino acids to the DFAA pool. These results support a unique profile of DOM derived from sponge exhalent seawater, with some components reduced in concentration and other components added or elevated in concentration, relative to the surrounding seawater.

The metabolic profiles of the exhalent seawater from the two sponge species were not significantly different and the overwhelming majority of mass features were present in the exhalent seawater from both species. Further work is needed to determine the extent to which DOM derived from other sponge species might be similar to our observations. Nevertheless, the degree of similarity of the exhalent seawater profiles between *I. campana* and *S. vesparium* was surprising given that these species have been characterized as HMA and LMA species, with different density and diversity of symbionts (*Poppell et al., 2014*). The dichotomy between HMA and LMA sponges is not always straightforward (*Gloeckner et al., 2014*), but several studies have documented functional differences relevant to biogeochemical cycling between the two categories of sponges (*Weisz et al., 2007*; *Freeman, Easson & Baker, 2014*). In the case of the species used in this study, there is a fairly well characterized dichotomy between the diverse and abundant community in *I. campana* (HMA) and the notably low density of microbes in *S. vesparium* (LMA; *Hardoim & Costa, 2014*; *Poppell et al., 2014*). While the exhalent seawater DOM profiles exhibited high overall similarity, more unique mass features were present in the exhalent seawater from *S. vesparium* than from *I. campana*. In addition, there was a higher number of features that were more abundant in the exhalent seawater of *S. vesparium* than of *I. campana* compared to the off reef samples. Further work is needed to ascertain the degree to which the overall DOM profile of seawater exiting from sponges represents general metabolic waste, either from the sponges or their microbial symbionts. Previous work with marine eukaryotic

phytoplankton observed differences in DOM exudate composition among phylogenetic groups (*Romera-Castillo et al., 2011*; *Becker et al., 2014*), consistent with the hypothesis that DOM exudate profiles could differ among closely related organisms such as sponge species as well as between different organism types. Future work should focus on whether the mass features observed in this study, particularly those unique to a sponge species or elevated in one of the two species, are the result of symbiont or host metabolism and whether such features are ecologically significant.

Both the targeted and untargeted metabolomics analyses suggest that sponges are reducing the concentrations of some DOM components and increasing the concentrations of others. This is expected because while the sponge holobiont would remove some compounds from the water for assimilation and/or energy, waste products from metabolism are also likely to be added to the seawater by the sponge. Consequently, sponge metabolism and filtration could have a significant impact on the pelagic microbial community and on the success of organisms that rely on DOM as a carbon source directly or indirectly *via* microbial symbionts. For example, boring sponges rely on dissolved organic carbon (DOC) to meet their energy demands (at least 80% of TOC they remove is DOC; *Mueller et al., 2014*). The impact of sponge-derived DOC, or DOM in general, on the metabolism of boring sponges has not yet been investigated. In fact, to our knowledge, there are no studies that have examined the influence of sponge-derived DOM on the growth and metabolism of any coral reef organism.

The ecological relevance of sponge-derived DOM in coral reef ecosystems would, in theory, be proportional to the impact that sponge metabolism has on the composition of DOM in the surrounding seawater. Of the 106 mass features that were higher in all of the inhalant seawater samples than the off reef samples, many ($n = 42$) were also higher in the exhalent relative to inhalant seawater. This suggests that the surrounding reef DOM profile may be partially derived from sponge exhalent seawater. Similarly, of the 71 features that were lower in the inhalant compared to off reef samples, 32 were significantly elevated in the exhalent samples relative to the inhalant samples, further distinguishing the inhalant and exhalent seawater samples from each other. The impact of these sponge-derived metabolites on coral reef nutrient dynamics is likely to be multifaceted, with individual compounds having different activity in the environment. Further experiments are necessary to determine the full impact of sponge-derived DOM on the reef DOM profile.

A portion of the local impact of sponge-derived DOM on seawater chemical composition, however, could be elucidated in the targeted metabolomics analysis. The targeted metabolomics dataset is a small subset of metabolites compared to the metabolic profile generated by the untargeted metabolomics analysis. Thus, it is not surprising to see differences between the two datasets. For example, the untargeted metabolomics analysis indicated a clear difference between the inhalant and exhalent seawater samples, while none of the quantified metabolites were significantly different in concentration between these two sample types. There was a general trend towards higher concentrations of some quantified metabolites, including several amino acids, nucleosides and the vitamin riboflavin ($B_2$) in the exhalent water relative to inhalant seawater, suggesting that sponges could be a source of these compounds in the reef ecosystem. Meanwhile, in comparison to

the reef samples, the two off reef samples contained higher concentrations of xanthosine, caffeine, and aromatic amino acids, indicating a non-sponge source for these compounds or rapid consumption of these compounds by the reef sponge holobiont. These preliminary data further support holobiont metabolism as a factor influencing reef DOM composition and suggest that, to some extent, there are differences in the composition of reef DOM relative to that of nearby off reef seawater.

Differences in metabolite concentration and composition between the reef samples and the off reef samples can be illustrated by caffeine (1,3,7-trimethylxanthine) and the purine nucleoside xanthosine. Caffeine exhibited similar behavior to the amino acids with high concentrations in the exhalant seawater relative to inhalant seawater, and particularly high concentrations in the off reef water compared to both the inhalant and exhalant seawater. Caffeine is a purine alkaloid found in plants and is considered a contaminant in freshwater and marine systems (*Weigel, Bester & Hühnerfuss, 2001*; *Gardinali & Zhao, 2002*; *Singh et al., 2010*; *Del Rey, Granek & Sylvester, 2012*), with the potential to negatively impact coral-algal symbiosis (*Pollack, Balazs & Ogunseitan, 2009*). Caffeine concentrations in the off reef samples were approximately 12 times higher than in the reef seawater even though the sponges examined here were a net source of caffeine. There is likely a non-sponge source for caffeine off reef given the concentration differences between on reef and off reef samples. Meanwhile, the nucleoside xanthosine was detected only in the off reef seawater samples, suggesting that either there is no major source of xanthosine within the reef environment or that there is a high uptake of xanthosine within the reef, resulting in a low standing concentration that may be below our detection levels.

The nearly constant source of DOM from sponges and the differences in DOM composition between the reef (inhalant and exhalant) and off reef samples may influence the planktonic community structure in these two habitats. In a four-year study in Mo'orea, such differences were observed in the composition and abundance of bacterioplankton community members between water overlying coral reefs and seawater away from the reef (*Nelson et al., 2011*). The presence of particular sponge species on the reef or overall sponge community composition could also influence reef DOC composition and concentration and, ultimately planktonic community structure. For instance, the Caribbean sponge, *Xestospongia muta* was found to feed selectively on live plankton, yet DOC, which comprised most of the TOC (84%) in the inhalant seawater, accounted for most of the sponge diet (70%) and the sponge retained DOC at rates of approximately −9 to 46%, exhibiting both release and uptake of DOC (*McMurray et al., 2016*). Although evidence exists that TOC/DOC concentrations are highly variable across sites within the Caribbean and across ocean basins (*Nelson et al., 2011*; *Haas et al., 2016*), no studies to date have investigated the potential link to sponge community structure. If there are inherent differences between locations in terms of carbon concentration and composition, future work characterizing the DOM profiles and the benthic communities of geographically disparate sites may help to link the flux of individual metabolites with specific members of these communities.

The fact that none of the metabolites driving the differences between sample types could be identified using metabolomics databases highlights the gaps in our knowledge of the composition of coral reef DOM and thus in the flow of biologically relevant nutrients in

this system. This is not surprising, given that metabolomics databases are still growing and are generally biased towards human-associated metabolites. Even recent human metabolomics studies, however, can only identify <2% of mass features generated by untargeted metabolomics analysis (e.g., *Bouslimani et al., 2015*; *Quinn et al., 2016*).

Classic examples of the impact of sponge holobiont metabolism on seawater chemistry include the addition of inorganic nitrogen and removal of DOC. Previous studies addressing such phenomena have demonstrated the complex roles that sponges play in coral reef ecosystems. The results presented here build on our understanding of the impact of sponges on the composition of coral reef DOM by demonstrating that sponges remove certain components of low molecular weight DOM and add other components. These actions resulted in a distinct profile of DOM in the exhalent seawater, which may influence the metabolic activity of the nearby pelagic microbial community, with potential implications for coral reef biogeochemical cycling. In areas with a high density of sponges, such as many of the off shore reefs in the Caribbean, sponge-derived DOM may have a greater impact on the overall profile of coral reef DOM. Additionally, the similarity of DOM profiles from sponges with distinct prokaryotic communities provides preliminary evidence for a common sponge signal in seawater DOM. While this present study was limited in size, it is the first high-resolution assessment of the impacts of sponge filtration and holobiont metabolism on coral reef DOM, an important reservoir of organic nutrients in this ecosystem. Finally, this work establishes a foundation for future hypothesis-driven studies aimed at better understanding the role of sponges in ecological and biogeochemical processes on coral reefs.

## ACKNOWLEDGEMENTS

The authors would like to thank Kenan Matterson and Cole Easson for assistance in the field, Michael Lesser for syringes used in sampling, Krista Longnecker for TOC analyses and Melissa Kido Soule for help with metabolomics analyses, and K Longnecker, M Kido Soule, and Winn Johnson for comments that improved an earlier version of this manuscript. The authors would also like to thank Krin Grande for artwork assistance. The manuscript was improved by constructive reviews from L Aluwihare and C Nelson. This represents contribution number 1047 at the Smithsonian Marine Station in Fort Pierce, Florida.

### Funding

This work was funded by 54 backers through the crowdfunding platform Experiment (https://experiment.com/projects/how-do-sponges-influence-the-availability-of-nutrients-on-coral-reefs). The funders had no role in study design, data collection and analysis, decision to publish, or preparation of the manuscript.

### Grant Disclosures

The following grant information was disclosed by the authors:
Experiment (54 backers).

## Competing Interests

The authors declare there are no competing interests.

## Author Contributions

- Cara L. Fiore and Elizabeth B. Kujawinski conceived and designed the experiments, performed the experiments, analyzed the data, contributed reagents/materials/analysis tools, wrote the paper, prepared figures and/or tables, reviewed drafts of the paper.
- Christopher J. Freeman performed the experiments, analyzed the data, contributed reagents/materials/analysis tools, wrote the paper, prepared figures and/or tables, reviewed drafts of the paper.

## Data Availability

 MetaboLights: http://www.ebi.ac.uk/metabolights/MTBLS281.

## Supplemental Information

Supplemental information for this article can be found online at http://dx.doi.org/10.7717/peerj.2870#supplemental-information.

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
