# Peer review of "Sponge exhalent seawater contains a unique chemical profile of dissolved organic matter"

_PeerJ, doi:10.7717/peerj.2870_

## Round 0.1 · original submission · Major Revisions

Please address all the comments of the reviewers in a point-by-point manner. In addition, I have the following comments:

Fig. 1: It appears that sampling was different for the two sponge species, with inhalant and exhalant samples taken from the same sponge for I. campana but not for Speciospongia -- is this correct?

Fig. 2 and L. 274: Which TOC samples were retained and contributed to Fig 2? From which sponge?

McMurray et al 2016 recently demonstrated that ~70% of the diet of X. muta is DOC. The authors need to clearly indicate that they did not quantify DOC and have limited data on TOC (and from which sponges?). Rather than a "characteristic DOM profile released," isn't it likely that selective uptake of DOM, with potentially considerable reduction on DOC concentrations, results in a different profile? This is conceptually different from sponges "releasing" DOM -- they may be differentially retaining DOM.

·

Basic reporting

INTRODUCTION
Lines 67-71. These two sentences interrupt the flow of the introduction and are not particularly meaningful. For example, it is not prudent to use concentration as a proxy for composition.

Lines 79-84. I recommend simply stating that "our understanding of the composition of coral reef DOM has been limited by the use of methods that fail to resolve the complex nature of DOM." The remaining discussion about LMW DOM is unnecessary. It can simply be stated that accessing different reservoirs of DOM by implementing different analytical approaches is necessary. It is also true that the statement that LMW DOM is a small component of total DOM is false (given the cutoff of 1000 Da) and furthermore, the assumption that LMW DOM is labile is also inconsistent with the DOM literature (especially that of Amon and Benner, the cited authors). My interpretation is that the authors are primarily referring to small, likely genetically encoded, metabolites rather than the more general fraction of DOM that is usually referred to as LMW DOM. If this is what the authors mean then they should be explicit about it. LMW DOM is not the appropriate term for the fraction they anticipate is a small component of total DOM and highly labile, especially when referring to the 1000 MW cutoff and Amon & Benner.

Line 155 – extra “filters”

RESULTS
The overall finding of the non-targeted analysis was the addition of peaks to the exhalant DOM – i.e. DOM exiting the sponge was significantly different from DOM entering the sponge. The targeted analysis was able to identify the addition of nucleosides in exhalant samples

Line 285. Is there no mention of PPL extraction efficiency? This is a necessary part of the methods section since it can influence the approach others might take. If the authors were unable to mass balance the C then they should state that.

Line 321. “Metabolites were generally highest in off-reef samples.” This is not exactly true. Six of the 15 compounds were more abundant.

Also, something is wrong with this plot – in the text caffeine is identified as being enhanced in off reef samples while Pantothenic acid is not. However, the plot shows something different. Also, would be worth noting in the legend that even though Xanthosine is discussed in the text, it does not appear in the figure because it was absent from reef samples.

Discussion
Line 343. Although distinct microbial communities are harbored by each sponge, it is possible that the exhalant “signature” is just one of generic intracellular metabolites (e.g., all organisms have nucleosides). Thus, it is perhaps premature to ascribe those signatures as a "sponge" signature. Perhaps I am misunderstanding the use of sponge here – do you mean the holobiont or the actual animal?

Line 348. I was confused by this statement at first. Essentially: because chemical composition is similar across samples the observation that exhalant samples are statistically different from both inhalant samples and off-reef water cannot be driven by compound diversity, and so, must be driven by differences in compound abundance/concentration. Perhaps use “that differences in the concentration of individual compounds” rather than simply “concentration differences”

I was additionally confused by this discussion because Figure 3 shows that ~340 compounds are unique to the Exhalant. Why don’t those compounds contribute to the observed statistical separation. I realize 340 mass peaks is a small percent of the ~2000 total. However, in the text where concentration differences are being discussed a reference is made to ~400 compounds being responsible (a number that is not much bigger than 340). It is possible that I am comparing apples and oranges here.

Line 394. How MANY of the peaks unique to reef samples (inhalant and exhalant) were found in the exhalant at higher concentration?

Line 449 This is a common problem for DOM from any environment.

Line 469”there is limited influence of sponge holobiont metabolism on the overall profile of coral reef DOM.” The authors should be explicit about what they mean here. Is it because only a small % of the peaks from the exhalant sample are enhanced relative to inhalant seawater?
There are several statements throughout the discussion (e.g. Lines 416-418; Line 458) that suggest a sponge influences is detectable. How much influence is "significant?"

Table 1. The “Significantly Different” row is confusing and should be explained better in the table legend. It is mostly confusing because the previous numbers are provided on a species by species basis whereas the numbers in this row represent pooled MS data from both sponge species. Also, why don't the data discussed in lines 299-304 appear in this table, when, as far as I can tell, these data should fall in the “significantly different” category?

Figure 2. Not necessary to be shown as a figure. Data are already described in the text.

Figure 4. I think it would be easier to immediately recognize the groupings if the colors for IC E and SV E were identical but their symbol shapes were different. Also, perhaps separate (e.g. hyphen) the E and I from C and SV or italicize the E and I.

Figure 5. Is this necessary? Seems like it would be pretty easy to summarize the data by inserting the total number of mass peaks in IC and SV, respectively, around Line 310-311

Experimental design

I want to commend the authors on the level of detail they provided in the methods section. They were unusually thorough.
That said, I was disappointed that there was no statement about the analytical bias associated with the extraction technique (and of course, the LC-MS analysis technique). At a minimum the authors should have noted the extraction efficiency of PPL cartridges for the reef and off-reef DOM. It is crucial to know how much of the total DOM was ultimately analyzed in the study because that affects the relevance of the findings reported here.

Validity of the findings

In general the conclusions reported in the study were supported by the data. For example, several statistical test were included to insure that significant differences could be identified as such. However, there were instances where some general statements were made without explicit reference to the subset of data supporting those statements. I have provided specific comments in the Basic Reporting section.

·

Basic reporting

OK. see above.

Experimental design

OK. see above

Validity of the findings

OK. see above

Additional comments

Overall, with some statistical caveats that may change some of the interpretations, this is a straightforward study with clear results. There are plenty of limitations to the data, but the key results, summarized cleanly at the start of the discussion and in the abstract, seem robust and informative.

That said, there is a lot to improve in this paper, and I have provided extensive recommendations because so many things are muddled and presented without clarity. I am confident that with careful attention to all of these points this manuscript can be acceptable for publication in PeerJ.

Abstract: I think that a brief summary of the targeted metabolites that were enriched inhalent or exhalent would be a good addition to the abstract and could replace the sentence about TOC/DOC which is uninformative and not a major point.

L58: please change to "...CAN have a major role..."

L64: "at a bulk level" is jargonistic. Perhaps "at an elemental level" or "in terms of total inorganic and/or organic elemental composition". In addition, L66-67 ("DOM includes" should be "DOM comprises") should be moved to midway L63 to preface the compositional sentence.

L82 "LMW fraction of DOM CAN BE a small component"

L83 change "are likely" to "have been shown"

L102: remove "unit"

L107: I would add that this is unsurprising, as it has been shown that many marine microbes are reluctant to incorporate glucose (Nelson and Carlson 2012).

L131: Nalgene is a brand, not a material. Define the plastic of the syringe and the bags, and specify if the large latex/rubber pieces were associated with the syringe. A part number would help as well. It is all well and good to acid wash things, and to use teflon tubing, but if you acidify certain plastics they leach more, not less, and certainly the rubber syringe heads leach considerable material.

L150: Again, define the plastic.

L154: Please indicate if the PTFE filters were pre-flushed to remove plasticizers common to teflon filters.

L159-161: This makes no sense. The method is called "TOC analysis" simply because it oxidizes all carbon with temperature, but you filtered the water to remove particles, so you should interpret the resulting data as DOC concentrations (DOC is operationally defined by the GF/F filter). L161 is confusing and defensive for no reason and should be removed because you already filtered the water.

L169-170: is it burdensome to briefly summarize the modifications of Longnecker (2015)?

L170: Please refer to this as DOM rather than "Metabolites"; the latter is a functional definition with little support based on your sampling strategy of collecting water (rather than tissue).

L179: Please clarify and summarize the standards used. If they were provided from Miami, how were they made fresh daily at WHOI? Most likely you used "reference waters" shipped from Hansell (which ones?) and "standards" made fresh daily locally around some range of concentrations?

L195: Clarify that the "pooled sample" is a mix of all environmental samples and not model compounds.

L233: ANOSIM would test compositional differences, but it is not clear what you mean by "abundance"; I recommend just removing that word unless you have an explicit intention for it.

L235: You need to change your statistic. Inhalent and Exhalent samples are "paired" or non-independent, and the appropriate nonparametric test is a Wilcoxon Signed Rank Test. You should justify why you are not using parametric statistics; it should be straightforward to transform your peak concentrations to be gaussian distributed for true paired t-tests, which would be preferable. for the species comparisons the W rank sum test is appropriate because the species are independent.

L236: Clarify that these six replicates are 3 inhalent and 3 exhalent, and that you have no criterion which of the two have the missing sample? That is confusing and most likely statistically invalid without correcting for differences in variance using a modified t-test like Welch's. I don't know how you might account for this with a Wilcoxon test or if you can., another argument for using parametric statistics.

L240: I don't follow the use of the term "relative" here...wouldn't these be absolute concentrations?

L244: Which samples were pooled? The exhalent samples from two species (n=6?) and some ambient samples (n=2?)? This is confusing.

L245: Here you are switching to parametric statistics, implying that you should just use these throughout.

L246: The Shapiro-Wilk test, and most "tests of normality" are overly conservative. You don't indicate whether those tests were significant or not, and whether metabolites that failed the test were left unanalyzed by ANOVA. I recommend skipping this test and just clarifying that data for each metabolite were approximately normal, and if you needed to transform the data to achieve this.

L247: clarify that the "16 metabolites" are of the 23 metabolites examined (Line 188) for which you have standard curves. It is not clear what the ANOVA is testing? Are you trying to see if any of the mean concentrations of metabolites differ in a set of samples? What set of samples are being averaged? Influent or effluent or offshore, which species?

L249: So you pooled across species for the PERMANOVA?

L258: So I assume since you used log-transformed concentrations for the PERMANOVA and CAP analyses you should use that same transform for the t-tests above instead of Wilcoxon. Be consistent across analyses.

L258-259: I don't understand why you are using NMDS and ANOSIM to analyze the untargeted data but using CAP and PERMANOVA on the targeted data. These are conceptually similar analyses and there is no justification for switching: one has 2000+ parameters and another 16, but the data are still concentration-normalized peak heights. Pick one framework and stick to it.

L261 The amount released per day should be using the mean+/-SD of the DIFFERENCE in ex-in as the concentration value in the formula. Make sure this is the case in Table 2. This equation is simple enough to communicate inline text as stating that you are just multiplying the Ex-In concentration by average water flux (L s-1 L-1) of each sponge, then averaging for the Table 2 and providing ranges. It might be argued that presenting this as a range using the range of flux rates would be better than using an average given the high variability of flux rates you observed in L274-275

L274 Clarify that pumping rates are in units of L water s-1 L-1 sponge.

L278 First you must clarify that these tests were done per species, using paired tests (Wilconxon signed rank or paired t-tests, again I suggest the latter as being most appropriate above) so you should be providing p-values for each species, or use some kind of linear model to include species as a random effect in the model.

L281 It is likely that these were just not acidified.

L284 Present the number of samples used for TOC analysis BEFORE you present the means and statistics. At this point the sampling is confusing enough that it makes sense to provide a table with the number of samples used for each of the three analyses (TOC, untagrgeted, targeted) from each species inhalent and exhalent and offshore. Or just indicate to look at the figure for the numbers of samples. I would recommend annotating Figure 1 to show how many samples were collected from each sample area.

L287-289 and Figure 3 don't seem very interesting, ditto for Figure 5. Perhaps just say that some 60% (1639) were found in all samples using your criteria, with 73-350 peaks found in just 1 or 2 samples? In Figure 3 What is meant by "Sample type"? Species? Inhalent/Exhalent/Reef? These are not 6 replicates, since you collected three replicates of each from each species, right? So is a peak that is found in 3 of one species and 1 of another species used in this analysis? I'm not sure that this is a very useful analysis, so perhaps make Fig 3 supplemental material? See my next comment for a replacement for Fig 3.

L293-314. These analyses seem reasonable to me (see recommendations for using parametric t-tests instead however) but I recommend that Fig 3 instead be Van Krevelen diagrams for both Species Exhalents highlighting the features enriched or depleted relative to the inhalent samples for that species. Ideally this is done with heat map coloring according to the magnitude of enrichment.

L323: is this an effect of influent effluent by species or across species? Be much clearer about which samples are being used in general in your comparisons, as you often just use the word samples.

Figure 6: The approach to deriving the fold-change is very confusing. I understand the intent, but frankly I don't think this convoluted approach is useful. I recommend plotting the ratio for each species just to be clear about the data. There would be no error bars.

Figure 7: be clearer that each ratio (in:ex) is done on a single sponge, then the ratios are averaged and presented with standard deviation whiskers. The remaining text in the legend is confusing: it is hard to imagine that for at least Adenosine and Guanosine, which each sponge gave a ratio of something greater than 2, would still not show a mean PAIRED difference between inhalent and exhalent. This may be solved when appropriate paired t-tests are used as I recommend above, or even iwth Wilc. signed rank tests.

Figs 6-7: Generally we use log2 ratios for "fold change" for intuitiveness and to avoid scaling issues: log2(reef/inhalent). You are effectively using log10 ratios in your figure by plotting the ratio on a log scale, but it is harder to interpret the numbers than it would be with log2 ratios.

Table 1: This legend is very poorly worded. The 20X explanation is terrible both here and in the text, and I can't follow it. The second sentence in the legend also means almost nothing to me, and I have read this manuscript very carefully. Please reword.

L332-334: While I appreciate the attempt to be quantitative, and this is fine for a piece of data for a table, is there any way to put this number in context?

L357: I don't follow how you incorporated the 1 sponge m-2 factor? Was that in the methods? I thought all of the numbers were normalized to L sponge?

L439: Again, you were analyzing DOC and need not qualify your data. Your DOC concentrations were nearly double those of typical oceanic (open ocean) habitats, so are likely not comparable to a remote high island like Moorea. It is likely that Caribbean reefs, especially off continental shelves, are just higher in DOC and you should make an effort to compare your DOC values with other studies in your area or in other areas of the Caribbean. There is some Caribbean coral reef DOC data in the synthesis of Haas et al 2016.

---

## Round 0.2 · Minor Revisions

This manuscript has undergone significant changes from the original submission, in many cases improving the submission and adding new data. There remain some important issues and questions that need to be resolved:

My first concern is that the authors did not appropriately address the following point made during the initial review:

“L235: You need to change your statistic. Inhalent and Exhalent samples are "paired" or non-independent, and the appropriate nonparametric test is a Wilcoxon Signed Rank Test. You should justify why you are not using parametric statistics; it should be straightforward to transform your peak concentrations to be gaussian distributed for true paired t-tests, which would be preferable. for the species comparisons the W rank sum test is appropriate because the species are independent.

Sokal and Rohlf (1995, Biometry), define a paired comparison as one where the “…pair is the same individual tested twice or of two individuals with common experiences…” (pg 352). We cannot be certain that the parcel of water collected as the inhalant sample is the same parcel of water collected as the exhalent sample and thus we treated these two samples independently. However, we changed the test to T-tests for all comparisons as suggested and described below. With only 3 samples it is not possible to assess the distribution with these data, which is why we decided to error on the side of a non-parametric statistic. However, it is not uncommon to use a parametric test anyway and we have changed our statistic to a T-test. While the impact of transformation is not assessable, we used a log transformation as suggested below.”

Regardless of whether or not the parcel of water collected as the inhalant sample is the same, inhalant and exhalent water samples are not independent if they were collected from the same individual. From Zar (2010, Biostatistical Analysis), independence implies “that each datum in one sample is in no way associated with any specific datum in the other sample”. Inhalant and exhalant samples are associated and represent pairs of measurements from one individual; therefore the appropriate statistical treatment is a paired comparison (see further elaboration by Sokal and Rohlf (1995, Biometry), p. 352, below their definition cited above).

Please provide a paired analysis of the data, as is appropriate in this case.

I would also advise the authors to remove the TOC analysis from the ms. for the following three reasons:

1) It is not particularly relevant to the primary aim of investigating how sponges may change the composition of DOM.
2) TOC includes POC, which may confound any comparisons, especially those between HMA and LMA species that may differentially feed on these carbon pools. Indeed, this may explain why there was no significant uptake of TOC, suggesting that the sponges were not feeding (line 285); however, the lack of significance may also be an artifact of not appropriately treating inhalant and exhalent samples as non-independent pairs (see above).
3) the sample size for the final analysis is very small and ambiguous. Specifically, (paragraph 275) the authors initially collected 6 inhalant and 6 exhalent water samples (three each from two species). If 3 inhalant samples were lost and 1 was contaminated, I would expect a total of 2 inhalant samples for analysis (6-3-1), not 3 as described (line 282). Similarly, if 2 exhalent samples were removed due to contamination, I would expect a total of 4, not 3 as described (line 282).

Other points:

Line 49-50, 393: I can’t find this result: “no significant differences in the metabolic profiles of exhalent water between the two sponge species”. The only interspecific comparison of metabolic profiles that I can find is paragraph 319, however it appears that abundance of compounds present in both IC-E and SV-E was tested (line 322). Why were the “substantial” numbers of unique compounds in the exhalent water of each species excluded from this analysis? Were the differences between complete exhalent metabolic profiles for the two species tested?

Line 263: since a volume normalized pumping rate is not used in this calculation, mean sizes of the two species measured should be reported in the results.

Line 355, main conclusion #3: see point above. Also, lines 315 and 321 are contradictory to this.

Line 378: check “20 pmol” against line 244 (2 pmol) and table 3

Line 398: add Gloeckner citation to References

Line 432: but what is the number of mass features that are higher in exhalent water, but the same or lower in inhalent water?

Paragraph 430: This seems contradictory to the second main conclusion (line 354). If the influence of sponge-derived DOM on the DOM profile of surrounding seawater is localized close to the exhalent plume, isn’t the “impact” of sponge metabolism on the composition of DOM in surrounding seawater negligible?

Line 461: “caffeine concentrations are higher in exhalent water” is contradictory to the idea that high concentrations in off-reef samples are a result of a non-sponge source (lines 453-454)

Line 467: if sponges are a net source of caffeine (Table 3), how can sponge filtration explain lower concentrations on the reef?

Line 477: This is contradictory to the third main conclusion (line 356). If interspecific exhalent DOM profiles are similar, why would you expect sponge community composition to influence DOC composition?

Line 503: is there really an influence of sponge holobiont metabolism on the overall profile of coral reef DOM if the influence is localized (lines 354, 375)?

Figure 4: how were ratios calculated given that samples were not paired? Was the same approach used as in Figure 3?

Table 1: why weren't all comparisons performed (i.e. “n.d.”). For example, the exhalent profiles for the two species were not compared, yet a major conclusion of the ms. is that exhalent profiles are similar between species.

Table 3: the legend for this table is exactly the same as the legend for Figure 3. Also, why are only a portion of the metabolites considered included in this Table?

---

## Round 0.3 · Minor Revisions

The major revisions from the last iteration were well done and this paper is close to ready. One last problem: although the authors have now addressed the limitations of their results in the discussion, the abstract was not revised to reflect this. Specifically, the abstract still states that (line 47) " These data suggest that sponges can influence the composition of DOM in the overlying water", which is misleading given that the results and discussion (e.g. lines 367 - 371) indicate the impact of exhalent seawater is minimal and localized to the sponge. Once the abstract has been revised to reflect the results and discussion, this paper will be ready to go.

---

## Round 0.4 · accepted · Accept

I expect this to be an often-cited paper on the leading edge of a topic of considerable future interest among marine ecologists.